# Documenting the implementation processes and effects of the data use initiatives in primary health care settings in Tanzania: A before-after mixed methods study protocol

Albino Kalolo[1,2]*, Anosisye M. Kesale[3], Mackfallen Anasel[3], Ntuli A. Kapologwe[4], Jennie Jaribu[5], Stella Mujaya[6], James T. Kengia[4]

1 Department of Public Health, St Francis University college of Health and Allied Sciences, Ifakara, Tanzania, 2 Centre for Reforms, Innovation, Health Policies and Implementation Research (CeRIHI), Dodoma, Tanzania, 3 School of Public Administration and management Mzumbe University, Morogoro, Tanzania, 4 Department of Health, Social welfare and Nutrition Services, President's Office Regional Administration and Local Government (PORALG), Dodoma, Tanzania, 5 UNICEF Tanzania, Dar es Salaam, Tanzania, 6 Data For Implementation (Data.FI), Palladium, Tanzania

* kaloloa@gmail.com

**Data Availability Statement:** No datasets were generated or analysed during the current study. All

## Abstract

### Background

The use of data in decision making and planning in primary health care settings is critical for improving efficiency and health outcomes for patients and communities. Implementation research can be used to fully understand the effects, context, challenges, and facilitators of data use, as well as how to scale up data use interventions. However, in the context of low resource settings, little is known about how implementation research can be employed to assess the implementation and impact of data use interventions.

### Methods

We will conduct a hybrid type 2 effectiveness-implementation study employing a mixed method controlled before and after design to measure the effects of data use interventions while simultaneously understanding the implementation of those initiatives. The controlled before and after entails measurement of the effects of the interventions at baseline and end line in a matched intervention and control health facilities using structured questionnaire to health workers (n = 440) and existing patients (n = 422) while also extracting selected health outcome variable from routine data in all participating health facilities (n = 80). The mixed methods component entails measuring the implementation outcomes (adoption, acceptability, fidelity and maintenance) and their moderators entails the integration of both quantitative and qualitative data collection, analysis, and interpretation (i.e. mixed methods) approach by using a structured questionnaire to implementers (health workers and managers) (n = 400). Experiential dimensions of implementation processes and moderators will be explored using qualitative interviews. Guided by implementation research theories and frameworks, a theory of change (TOC) is developed first to guide the evaluation of implementation

relevant data from this study will be made available upon study completion.

**Funding:** This work is supported The United Nations Children's Fund (UNICEF) through a grant to the Centre for Reforms, Innovation, Health Policies and Implementation Research (CeRIHI). UNICEF provided funds for data gathering and analysis but has no role in the design, analysis and interpretation of the results from the proposed study.

**Competing interests:** Jennie Jaribu works with UNICEF, Tanzania. Other authors declare no competing interests .This does not alter our adherence to PLOS ONE policies on sharing data and materials.

**Abbreviations:** CERIHI, Center for Reforms, Health Policies and Implementation Research; CFRI, Consolidated Framework for Implementation Research, CHMT: Council Health management Team; COM-B, Capability, Opportunity, Motivation, Behavior; DHFF, Direct Health Facility Financing; DHIS2, District Health Information System 2ANC; HFGCs, Health Facility governing committee; HIMS, Health Management Information Systems; HIV, Human Immunodeficiency Virus; IDI, In-depth Interviews; LMIC, Low and Middle income countries; MOH, Ministry of Health; NaTHREC, National Health Research Ethics Committee; PHC, Primary Health Care; PORALG, President's Office—Regional Administration and Local Government; PRISM, Performance of Routine Information Systems Management; REAIM, Reach, Effectiveness, Adoption Implementation Maintenance; RHIS, Routine Health Information Systems; RMNCAH, Reproductive, Maternal, Neonatal, Child and Adolescent Health; SDG, Sustainable Development Goals; SOPs, Standard operating procedures; TOC, Theory of Change; UHC, Universal Health Coverage; UNICEF, United Nations Children's Fund; USAID, United States Agency for International Development; WHO, World Health Organization.

processes and effects of the interventions. Descriptive and inferential statistics will be employed to analyze quantitative data whereas thematic analysis approach will be employed for qualitative data.

## Discussion

This study is one of the first to test the simultaneous measurement of effects and implementation processes of data use interventions in the primary health care settings. Findings will support efforts to improve quality of services by optimizing scale up and sustainability of the data use initiatives in primary health care settings.

## Introduction

In an effort to achieve the sustainable development goals (SDGs), specifically so goal number 3: good health and wellbeing, low and middle income countries (LMICs) are striving to hasten the pace under which health system strengthening initiatives are implemented [1, 2]. There is growing evidence on the importance of using quality and timely data from the health management information systems (HIMS) to inform the progress of health system strengthening initiates across all the WHO health system building blocks [3–5]. To achieve the targeted health system strengthening goals, quality data is required for the planning and implementation of each of the health system building blocks [6].Existing evidence indicates that positive data experiences lead to a need for more data and a sustained commitment to improving data quality and data use [5, 7]. In addition, there is a relationship of improved information, demand for data, and continued data use and this contributes to improved performance and unexpandable benefits of health programs and policies [4, 5].

Existing evidence also attests to the fact that few health facilities use data collected through the routine HIMS for decision making and planning [8–10]. While the use of data at health facility level has benefits in data driven decision making such as in budget preparations, staffing decisions, medical supply and planning clinical services, several factors have been implicated to influence it. The factors reported in literature include: governance (leadership, participatory monitoring, regular review of data and supervision) [9, 11, 12], type (level) of a health facility [13]; production of information (data analysis ,presentation of results, and data quality); health systems inputs (human workforce, financial resources) [9, 13] and health information system resources (electronic health management information systems, organizational structure, training) [11]. Proactive interventions that promote use of data in primary health care settings may hasten use of data for decision making and planning and therefore help to strengthen health systems. As such, data use initiatives that breakdown barriers to data use, improve availability of ready to use data and build capacity of health workforce to effectively use data may help to improve quality of care, strengthen health systems and hasten achievement of universal health coverage (UHC).

Based on the PRISM (*Performance of Routine Information Systems Management*) framework, for data use intervention to be effective, they should address inputs, processes and outputs or performance of routine health information systems (RHIS) [14]. The PRISM framework defined RHIS performance as better quality data and continuous use of information. RHIS performance is a function of RHIS processes and their technical, behavioral and organizational determinants. Addressing these determinants may lead to improved quality of care, better health outcomes and system performance.

In Tanzania, UNICEF and the Data For Implementation (Data.FI) project, funded by the United States Agency for International Development (USAID), are implementing a data use initiative in the primary health care settings with a goal of improving data use in decision making and planning at different levels, Data.FI providing support at the council level while UNICEF in collaboration with PORALG providing support at the facility level. The interventions are implemented in the two regions of Tanzania namely Dar es Salaam and Dodoma, in which two local government authorities (councils) in each of the region have been recruited in the intervention.

Interventions to address data use in primary health care settings are often complex with multiple components. Complex interventions require extensive stakeholder involvement, longer time frames, and are subject to strong contextual influences [15, 16]. As a result, they pose evaluation design and implementation challenges, necessitating a comprehensive approach to program evaluation and monitoring. Desirable evaluation designs are those that aim at: 1) understanding the implementation processes of the intervention and the moderating factors, specifically describing in detail the adoption, fidelity program reach, beneficiaries/client responsiveness and quality of implementation of key process indicators as well as sustainability indictors 2) Using the implementation outcomes to explain the probable variations of effects observed when measuring the impact of the program.

Intervention implementation strategies targeting data use determinants can improve implementation outcomes, such as adoption, acceptability, fidelity and sustainability. Supportive supervision and technical backstopping designed to overcome implementation barriers and leverage innovations to foster data use initiative implementation represent promising approaches to facilitate implementation [11, 17, 18].

Documenting and assessing program implementation processes, program effects, and understanding contextual factors that may affect both the implementation process and program effects in the context of health system strengthening is a critical component of any robust and comprehensive evaluation effort of complex health system interventions implemented in real-world settings. Studies that explicitly link implementation processes to program outcomes have been found to be informative when taking interventions into scale or implementing in other settings as it is easier to draw lessons [10, 11].There is a litany of evidence that the level and process of implementation affect the outcomes of a program [18–21]. It follows that assessing the implementation process of interventions offers potentials to describe the mechanisms through which a given intervention produces outcomes, thus documenting both expected and unexpected effects.

Finding from such evaluations contribute to continuous learning and lesson drawing across settings in the stride forwards to strengthen health systems as countries strive to move towards universal health coverage (UHC). In the data use domain, little is known on how the interventions are implemented, the effectiveness of the intervention and the influence of contextual factors in relation both to implementation and outcomes of the interventions.

The objective of this study is to disentangle the implementation processes of data use initiatives (adoption, acceptability, fidelity, maintenance, and cost), understand the determinants of implementation outcomes, and determine the effects of data use initiatives on quality of care and selected health outcomes.

In this protocol, we aim to unravel the implementation processes (adoption, acceptability, fidelity, maintenance and cost) of the data use initiatives (interventions that focus on promoting data use culture by improving data availability and usability, facilitating use of data by health workers for decision making and planning (routine and long-term) and breaking down barriers to data use) , understand the determinants of implementation outcomes and determine the effects of data use initiatives on quality of care and selected health outcomes

## Materials and methods

### 1. Study settings

Intervention and comparison districts will be purposively selected from Dar es Salaam & Dodoma and Tanga & Mwanza Regions respectively with a view to minimizing contamination between intervention and comparison arms. In each region, the study area will include 20 health facilities, i.e., 10 health facilities from each district council. Health facility surveys will be conducted at baseline and end line in both the intervention and comparison areas, whereas the data use intervention will reach only the intervention district councils (two districts each in Dar es Salaam (Temeke and Kinondoni) and Dodoma (Dodoma City Council & Chamwino). The intervention districts will also be followed up at 6 months to measure the implementation outcomes and their determinants. To contextualize the settings of this study, we describe the study sites and the program.

**a. Study sites.**   The intervention districts will be selected purposively based on the fact that the data use interventions will be implemented. The control districts will be selected from two regions to match the intervention districts based on geographical location (urban vs. rural). The health facilities from the selected districts were selected based on the following criteria: a primary public health facility, providing services for more than a year. Hereunder the characteristics of the selected districts are described.

Kinondoni and Temeke are urban districts (Municipality) in the Dar es Salaam Region, the business capital of Tanzania Mainland. The census of 2012 showed that the population of Kinondoni was 1,775,049 whereas that of Temeke District is 1,368,881.

Chamwino and Dodoma City councils are districts in Dodoma Region, the administrative capital of the united republic of Tanzania. The census of 2012 showed that the population of Chamwino was 330,543, whereas that of Dodoma city council was 410,956.

Tanga city council and Korogwe are districts in the Tanga region, one of the important cities in the coastal zone of the country. The census of 2012 showed that the population of Korogwe was 56,282 whereas that of Tanga city council was 273,332.

Mwanza city council and Misungwi District council are districts in Mwanza region, one of the important cities in the Lake zone region of the country. The census of 2012 showed that the population of Mwanza city council was 706,453 whereas that of Misungwi was 351,607.

The eight councils represents a mixture of metropolitan (Temeke & Kinondoni), urban (Dodoma City council, Mwanza city council, Tanga city council) and rural district councils (Chamwino, Korogwe and Misungwi). The study areas are suitable of generating outcomes that could easily be transferable to other similar settings in the country.

**b. Description of the data use initiatives.**   This implementation study is conducted in the context of an intervention that is set to promote data use for decision making and planning to improve the quality of care and health status (outcomes) of the population.

The President's Office—Regional Administration and Local Government (PO-RALG) requested assistance from UNICEF and USAID to promote and institutionalize a culture of data use at subnational level in the stride forwards to achieve universal health coverage (UHC) and strengthen PHC systems in line with the decentralization policy and related approaches such as the direct health facility financing (DHFF) . USAID engaged the Data for Implementation (Data.FI) project, to support the implementation of the request with the aim of using quality data to support evidence-based health programming and accountability.

In April 2021, UNICEF and Data.FI drafted a proposed joint implementation plan to support implementation of data use for this quality improvement initiative at different levels of the health care system, with Data.FI, providing support at national, regional and council level whereas UNICEF supports the health facility level. The intervention strategies aim to address 1)

organizational determinants of data use (governance, equipment & tools and planning ) 2) technical determinants of data use (physical infrastructure, ICT infrastructure , HIS design and operability 3) behavioral determinants of data use (capability, opportunity and Motivation )

## 2. Study design

This study will adopt an effectiveness-implementation type-2 hybrid design [22] to evaluate the effectiveness of the data use initiatives in improving quality of health service delivery and health outcomes while simultaneously understanding the implementation of the initiatives. Since the data use initiative is implemented in twenty health facilities of two regions, we plan to carry out a before and after controlled design to estimate the combined effects of the data use initiatives in the intervention sites and matched control regions. Simultaneously we plan to assess the implementation processes of the initiatives and this entails the integration of both quantitative and qualitative data collection, analysis, and interpretation methods (i.e. mixed methods) to understand the implementation processes and the influencing factors.

## 3. Conceptual frameworks

In order to assess the effects, implementation outcomes and influencers of the data use initiative in the primary health care settings we use a conceptual framework that acknowledges that improving health information systems [23] in the primary health care settings can improved data use and therefore strengthen the primary health care, improve quality of care and health outcomes (see Fig 1).In addition, we acknowledge the contribution of the capability, opportunity, motivation and behavior (COM-B) model in changing behaviors of organizations and individuals to be favorable for data use in the primary health care settings. Our conceptualization is also inspired the implementation science frameworks that include: 1) Conceptual Framework for Implementation Outcomes [24], in this study, we consider all the three outcomes of implementation research namely, implementation outcomes, service outcomes and client outcomes 2) Reach, Effectiveness Adoption Implementation Maintenance (RE-AIM) framework [25] 3) Consolidated Framework for Implementation Research (CFIR) [26]. Additional guidance for specific constructs are derived from the fidelity of implementation framework [18], theoretical framework for acceptability [27].

## 4. Study components

The steps to conduct this study will be based on the two main components of the study, that is , effectiveness and implementation process .The effectiveness component entails collecting data to measure outcomes of the initiatives on selected indicators (top level indicators for service areas such as maternal, newborn, adolescents and child health ) at PHC before and one year after the implementation of the program .The implementation process study aims at understanding how the program is implemented amid contextual influences. To complete the components, the study will adopt the following steps.

**1) Develop a theory of change (TOC) for data use initiatives and document the process under which such a TOC is developed.** In order to understand the mechanism through which outcomes of data use initiatives are produced, we follow the implementation science steps throughout the conduct of this study. The steps are 1) developing the theory of change (TOC by reviewing internal program documents, attendance to various meetings on the data use initiatives and consultations with program developers and implementation teams and literature review on similar programs targeting primary health care settings 2) Testing the TOC, by developing tools and collecting empirical data in the field 3) Refining the TOC through a qualitative work consisting of in-depth interviews with key stakeholders involved in, or who

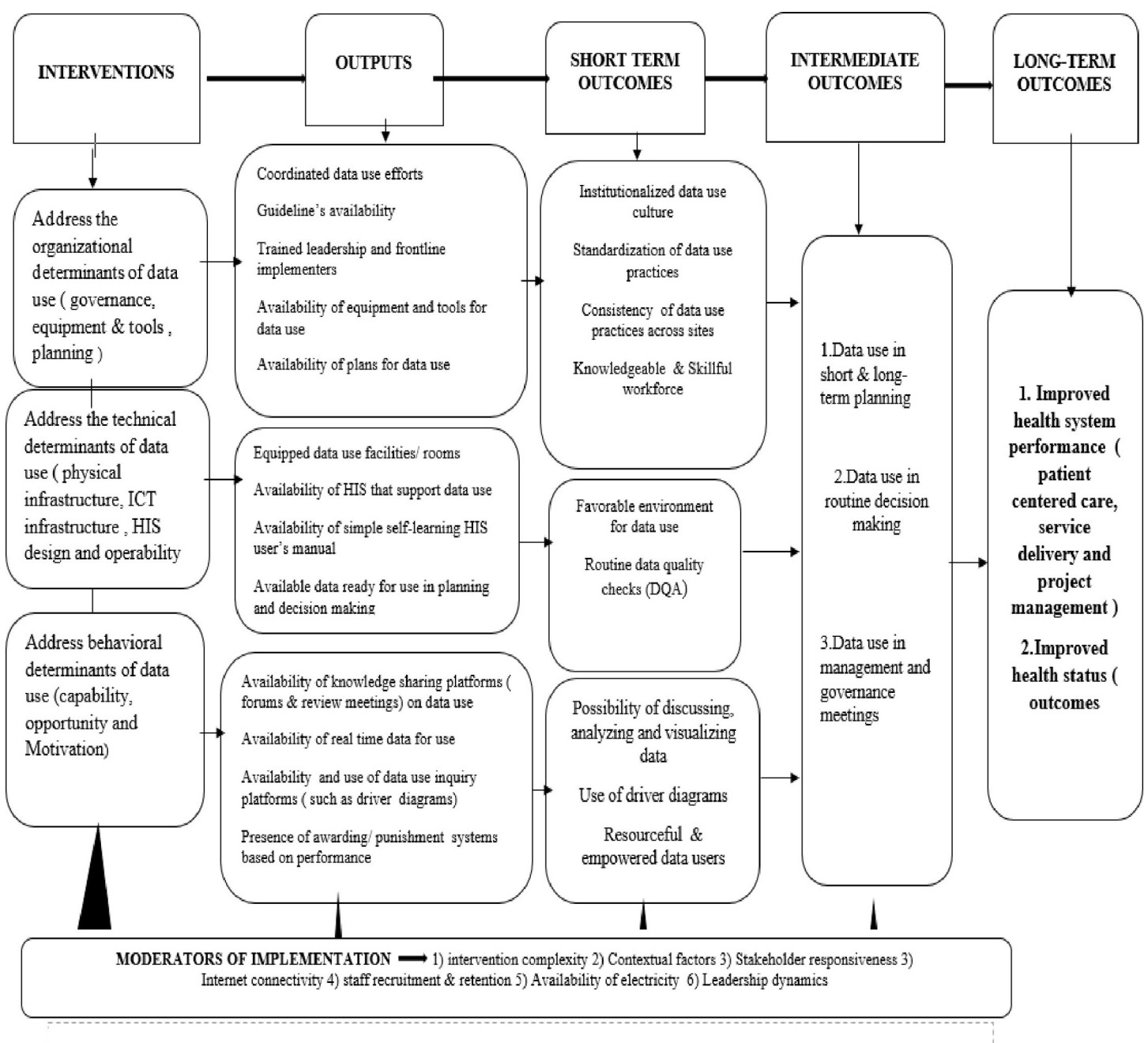

**Fig 1. Theory of change for data use in primary health care.**

have a stake in data use initiative, and focus groups and in-depth interviews with program beneficiaries, i.e., the health care mangers and frontline health care workers. The initial TOC has been developed (see Fig 1 & S1 File for a details) in order also to guide the development of data collection tools at all stages of the study.

**2) Understand the initial (baseline) situation of data use in the study settings in relation to decision making and comprehensive (council or health facility) planning.** To understand the baseline situation, the four councils implementing the data use initiative will be

matched equally with comparable councils where the data use interventions will not be implemented. Stratified sampling (to distinguish the level of facilities) will be used in these four councils to select a representative sample of facilities to be observed for one year after the project is implemented. This step combines quantitative and qualitative methodologies to evaluate the current state of data use in the recipient health facilities.

**3) Understand the implementation processes (acceptability, adoption, fidelity and sustainability) of the data use initiatives amid the influence of contextual factors.** Complex interventions are often criticized as being a black box because it is difficult to know exactly why an intervention did (or did not) work. To address this gap and open the black box, a process evaluation will be completed using a convergent parallel mixed methods design to assess acceptability, adoption, fidelity and maintenance of the intervention processes. The convergent parallel mixed methods component refers to the integration of both quantitative and qualitative data collection methods to complete the process evaluation at the same time. The process indicators reflect the immediate outputs of the data use initiatives. In line with existing evidence on intervention uptake and sustainability, the process evaluation will elucidate how the initiatives have been implemented in the intervention councils and why [21, 27–31]. In turn, this information will be used to explain the probable heterogeneity of effects observed when measuring the effect of the program. Triangulation of data from multiple sources will be done to generate comprehensive information from this study and generate very informative body of knowledge to inform data use related policies and guidelines in the primary health care settings. All the above activities will be guided by the initial TOC developed in this study.

Data will be collected from health managers and frontline workers delivering services in health facilities. Quantitative data collection will be done using structured questionnaire to health care managers and frontline health workers who produce data at multiple points and a document review checklist in order to assesses both the implementation outcomes (acceptability and fidelity) and service outcomes (efficiency, safety and patient centeredness).

The specific research questions in this component will be as follows:

1. To what extent the data use initiative has been adopted, accepted and maintained?

2. How many users of primary health facility have been reached by the initiative?

3. To what extent do the implementers and beneficiaries of the initiative adhere to the initiative protocol?

4. Is data use initiative faced with implementation challenges? What are those challenges and how can they be addressed?

To answer the above research questions, the following objectives will have to be addressed

1. To assess the extent of adoption of the data use initiatives

2. To determine number or proportion of beneficiaries reached by the data use initiative

3. To assess the fidelity to data use initiative protocol and SOPs

4. To assess acceptability of the data use initiative by primary beneficiaries and other stakeholders

5. To assess sustainability of the data use initiative in the intervened primary health facilities

6. To determine the implementation challenges and success of the data use initiatives

7. Understand the effects of the data use initiative on health system and patient outcomes

To understand the effects of the data use initiative, we will collect data at baseline and end line in order to allow two points measures of the outcome data. In this step, to assess compliance to national and international quality standards, the study team will rely both on the analysis of the routine health information system (at the facility level and district level) and information collected from respondents. Specific research questions include, i) has the quality of primary health facility data improved following implementation of the initiative? ii) What factors have contributed to the observed changes?

Specific objectives:-

1. Assess the effect/impact of data use initiative on beneficiaries' capabilities (knowledge, and skills to analyze and interpret data)

2. To evaluate the impact/effect of data use initiative on the quality of data collected in the primary health facilities

3. To determine the impact/effect of the initiative on the use of data for decision making and planning

4. To evaluate the effect of data use initiative on the performance of primary health facilities in terms of quality of delivered health services (availability drugs, medical supplies and equipment) and client's satisfaction with the services.

*Study component 1*: *Measuring effects of the data use initiative*. This component aims at establishing the effects of data use initiatives on quality of care and selected health outcomes. To establish and control the counterfactual, data will be collected in the intervention and control councils. This component comprises collection of quantitative and qualitative data at two points (baseline and end line) covering a period of one year.

*Sampling and sample sizes*. This component will collect information from health facility documents , health care workers, and patients. The samples and sample sizes for this quantitative component are as follows

1. We will collect health facility level data to assess the overall situation of quality of care from a total of 80 health facilities (40 from intervention health facilities & 40 from control health facilities).

2. From each of the selected health facilities we will collect quantitative data from at least five participants (the chairperson of the health facility governing committee, in charge of the health facility, data focal person , in charge of RMNCAH services and in charge of outpatient services) making a total of 400 participants. This sample is adequate to conduct robust inferential analyses related to the effects of the data use initiative. Moreover, taking p = 50% and applying Cochrane formula while also accounting for refusals and the fact that some health facilities (especially the dispensaries) may not all the five staff, an addition of 10% to the minimum sample will be added to reach 440 participants.

3. Also, from the selected facilities exiting patients will be systematically selected after gender stratification following medical consultations. The exit interview with Client/patients will be approached after they have received the services and are ready to go home. Respondents eligible for interview include all exiting patients or relatives of patients (aged above 18 years). They will be sampled to ensure equal numbers of men and women are captured. The sample is calculated using the Cochran formula (1977) [32]; by taking 50% as a proportion of patients' perception on quality of primary health care services) and a power of 80% allowing for an estimated error margin of 5%, the sample size obtained was 384 patients. To

account for refusals, an additional 10% of the sample will be added (n = 38), making a total of 422 patients.

The qualitative samples and sample sizes for this component are as follows:

We will use purposive sampling technique, to select at least 8 participants for in-depth interviews as a sub-sample of exit interview questionnaire (for exiting patients) from each district, i.e., 4 participants from dispensaries and four participants from health centers, thus making a total of 64 patients. Also, a total of 4 health care workers will purposively be selected for in-depth interview per council, i.e., 2 participants from dispensaries and 2 participants from health centers, thus making a total of 32 in-depth interviews. The exact number of interviews will be based on saturation of information that is the sample could even be smaller or increase depending on saturation. The interviews will explore the process and experiential dimension of quality of care and will complement evidence emerging from the quantitative analysis.

*Data collection tools and procedures*. To understand the data use situation including their determinants (governance, technical, behavioral and context) and existing initiatives and the indicators related to quality of care at the health facility, we will rely on facility assessment tool, interviews with clients and a structured questionnaire to health care workers and an observation checklist. The tools aim at getting a holistic picture on changes in quality of care and related health outcomes overtime as the initiative is implemented or as a result of changes in time. The tools are described here under and are found in S2 File.

1. Facility assessment tool: We will use the star rating tool to understand the baseline situation in relation to quality of care of the health facilities. The star rating tool is a validated instrument that has been used to measure quality of care in the health facilities in Tanzania. We will measure the changes in indicators related health system building block indicators (financing, availability of health commodities, and motivation of human resource for health) that could reflect the effects of data use at health facility level.

2. Patient exit interview questionnaire: We will use a patient exit questionnaire to understand patient's experiences and perceptions on the quality of care provided in health facilities. These questionnaires are also part of the star rating tool, i.e., we will adopt all the questionnaire in the star rating tool
NB: The star rating tool consists of facility assessment and exit interview questions and has been used repeatedly to assess quality of health care services in Tanzania [33]. During the baseline assessment, we will also pilot and validate an organizational readiness tool geared to assess health workers readiness to implement data use interventions

3. Health workers questionnaire: We will use a health workers questionnaire to understand the situation in relation to data use and the determinants of data use at the facility

4. Observation checklist: We will use an observation checklist that aims at getting the reality in relation to data use tools, equipment and practices related to data use at the health facility

5. For the health outcomes: we will extract data from DHIS2 for selected indicators using a document review checklist/DHS2 data extraction checklist. The indicators include: Number of health facility deliveries, number of pregnant mothers completed 4th ANC visit, NC Total Pregnant Women (PW) tested, Number of pregnant women who tested positive for HIV (First Test), Proportion of pregnant women on ARV to prevent HIV transmission and Number of Maternal Deaths

*Study component 2*: *Measuring implementation process of the data use initiatives*. This component aims to establish data use initiative implementation processes as well as influencing (moderating) factors. This component will be implemented in the intervention councils to better understand the initiative's implementation processes, thus unlocking the data use initiative's black box by better knowing the initiative's components and contextual impacts.

*Sampling and sample sizes*. Our sample for measuring the implementation process (process evaluation) will include implementers of the data use initiative at the council and health facility level. The participants will be implementing partners, health care mangers and health facility governing committee members (CHMT, HFGCs, health facility-in charges and frontline health workers). The fact that all CHMT members are guardians of data use at the council and its respective health facilities, they will all be included in the study (census) thus making a total of 60 CHMT members i.e., at least 15 members from each of the participating council. At least two members of HFGCs will be recruited in the study making a total of 80 HFGC members. A consecutive sample of health care workers from the participating health facilities with a maximum of five participants based on the assumption that the selected facilities should have a minimum of five health care workers, i.e., a total of 200 health workers will be recruited in the study. In addition, routine health facility files and documents will be read and summarized to fit the study context.

*Data collection tools and procedures*. The domains for the implementation process evaluation are reach, adoption, fidelity of implementation, acceptability and sustainability. Data will be collected from the following types of participants and documents; implementing partners, health care managers and health facility governing committee members (CHMT, HFGCs, health facility -in charges and frontline health workers) and health facility documents. Table 1 provides the details on the set of tools and their alignment to objectives and sources of data. In line with our mixed methods design, quantitative data will be gathered using; structured questionnaires, document review/ observation checklists, and semi-structured in-depth interviews (IDIs).

The questionnaire for health workers will capture information related to socio-demographics, general knowledge on data use initiative, acceptability of the initiative to health workers, fidelity of implementation, sustainability of data use initiative and moderating factors. The questions in this tool will either be multiple choice, yes/no or Likert scale.

A structured observation checklist will be used to collect data on implementation of data use initiative. This will serve as triangulation of the information obtained from the health care workers. We will use semi structured instruments to collect data from the implementing partners, health care mangers at all levels and frontline health workers. The qualitative component aims at uncovering the participant's experiences with data use and their expectations on the data use initiatives and its future maintenance at the health facility. All the tools to be used in this study are detailed in S2 File.

## 5. Analytical procedures

Descriptive analysis will be used to summarize data where by continuous data will be summarized using mean and standard deviation (SD) as well as medians and interquartile range. Dichotomous variables will be summarized using frequency and percentage. Chi-square test or fisher exact will be used to evaluate bivariate associations between categorical variables where by t-test will be used to estimate association in numeric variables and ascertain whether there mean different between categories. We will also use regression analysis.

Audio-recordings of in-depth interviews and focus group discussions will be transcribed verbatim in Kiswahili guided by a transcription guideline and translated to English.

**Table 1. The set of tools and their alignment to objectives and sources of data.**

| Study component | Objective | Domain/ Data required | Sources of data | Data collection tools |
|---|---|---|---|---|
| Development of a Theory of change | Develop a theory of change (TOC) for data use initiatives and document the process under which such a TOC is developed | | Documents, participants (data use initiative designers and implementers) | Document review checklist, Semi structured interview guides for in-depth interview |
| Baseline evaluation | Understand the initial (baseline) situation of data use in the study settings in relation to decision making and comprehensive (council or health facility) planning | Socio-demographic characteristics Governance for data use Data use behaviors Equipment and tools for data use Contextual factors for data use | Documents, participants (health facility managers, health care workers, district managers/ CHMT members) and members of Health facility governing committees | Document review checklist, Structured questionnaire, semi structured interview guides for in-depth interview |
| Process evaluation | 1) To determine the adoption of data use initiative by the health workers | Uptake of the data use initiative by the implementers (proportion and representativeness of individuals involved in implementing the initiative) [ADOPTION] | Documents, participants (health facility managers, health care workers, district managers/ CHMT members) and members of Health facility governing committees | Document review checklist, structured questionnaire, Semi structured interview guides for in-depth interview |
| | 2) To determine the reach of the data use initiative to health workers at the CHMT and the health facility level | The proportion of health care workers reached by the initiative [REACH] | Documents, participants (health facility managers, health care workers, district managers/ CHMT members) and members of Health facility governing committees | Document review checklist, structured questionnaire, Semi structured interview guides for in-depth interview |
| | 3) To assess the fidelity to data use initiative protocol and SOPs | The extent to which the implementers adhered to the original plan to implement all the essential elements of the program; any deviations or adaptations to the original plan, and follow-up activities [FIDELITY OF IMPLEMENTATION] | Documents, participants (health facility managers, health care workers, district managers/ CHMT members) and members of Health facility governing committees | Document review checklist, structured questionnaire, Semi structured interview guides for in-depth interview |
| | 4) To assess acceptability of the data use initiative by primary beneficiaries and other stakeholders | Information on affective attitude, burden, ethicality, intervention coherence, opportunity costs, perceived effectiveness and self-efficacy [ACCEPTABILITY] | Documents, participants (health facility managers, health care workers, district managers/ CHMT members) and members of Health facility governing committees | Document review checklist, structured questionnaire, semi structured interview guides for in-depth interview |
| | 5) To determine the extent to which the data use practices become part of the routine health facility practices and maintain its effectiveness | Maintenance/ sustainability | Documents, participants (health facility managers, health care workers, district managers/ CHMT members) and members of Health facility governing committees | Document review checklist, structured questionnaire, semi structured interview guides for in-depth interview and FGD |
| | 6) To determine the implementation challenges and success of the data use initiatives | Contextual factors | Documents, participants (health facility managers, health care workers, district managers/ CHMT members) and members of Health facility governing committees | Document review checklist, structured questionnaire, Semi structured interview guides for in-depth interview and FGD |
| Outcome evaluation | 1) Assess the effect/impact of data use initiative on beneficiaries' capabilities (knowledge, and skills to analyse and interpret data) | impact of data use (knowledge, skills) | Documents, participants (health facility managers, health care workers, district managers/ CHMT members) and members of Health facility governing committees | Document review checklist, structured questionnaire, Semi structured interview guides for in-depth interview |

*(Continued)*

**Table 1.** (Continued)

| Study component | Objective | Domain/ Data required | Sources of data | Data collection tools |
|---|---|---|---|---|
| | 2) To evaluate the impact/effect of data use initiative on the quality of data collected in the primary health facilities | impact of data quality (relevance; accuracy; credibility; timeliness; accessibility; interpretability; and coherence) | Documents, participants (health facility managers, health care workers, district managers/ CHMT members) and members of Health facility governing committees | Document review checklist, structured questionnaire, semi structured interview guides for in-depth interview |
| | 3) To determine the impact/effect of the initiative on the use of data for decision making and planning | | Documents, participants (health facility managers, health care workers, district managers/ CHMT members) and members of Health facility governing committees | Document review checklist, structured questionnaire, semi structured interview guides for in-depth interview and FGD |
| | 4) To evaluate the effect of data use initiative on the performance of primary health facilities in terms of quality of delivered health services (availability drugs, medical supplies and equipment) and client's satisfaction with the services. | availability of health system inputs, health system outcomes (responsiveness, quality of care and utilization of services) and client satisfaction | Documents, participants (health facility managers, health care workers, district managers/ CHMT members) and members of Health facility governing committees | Document review checklist, structured questionnaire, Semi structured interview guides for in-depth interview |
| | 5) To evaluate the effect of data use initiative health outcomes | Patient pathways, patient outcomes | Documents, participants (health facility managers, health care workers, district managers/ CHMT members) and members of Health facility governing committees | DHIS extraction form, Document review checklist, structured questionnaire, semi structured interview guides for in-depth interview |
| | 6) To determine the factors influencing the observed change in outcomes | Contextual factors | Documents, participants (health facility managers, health care workers, district managers/ CHMT members) and members of Health facility governing committees | Document review checklist, Semi structured interview guides for in-depth interview |

Transcription and translations will be done by the research team that conducted the respective IDIs. Codes will be grouped into categories, sub-themes and themes. Thematic analysis will be used to analyze qualitative data using ATLAS.ti 8 software. Table 2 provides details on data collection tools, administration, sample size and analytical techniques.

## 6. Ethics approval and consent to participate

This protocol explains a study that is being done as part of the data utilization initiative. The protocol received ethical clearance certificate from National Health Research Ethics Committee (NaTHREC),Ref. NIMR/HQ/R.8a/Vol.IX/3940.Permission to collect data in the regions and councils will be sought from the President's Office Regional Administration and Local Government, the Ministry of Health(MOH), the Regional Secretariat (RS), and the Local Government Authorities (LGAs). All study participants will be asked to provide written informed consent. In publications resulting from this study, no personally identifying information will be disclosed.

## 7. Dissemination

The results from the current study will be shared with participating districts and health facilities, disseminated through scientific conference presentation and peer reviewed journals.

**Table 2. Data collection tools, administration, sample size and analytical techniques.**

| S/N | Data collection tool | Level (s) of administration | Method of administration | Sample size (effects measurement) | Sample size (process evaluation) | Analytical techniques |
|---|---|---|---|---|---|---|
| 1 | Document review checklist | CHMT and Health facility level | physical and e-records, in ODK | Health facilities (N = 40), Districts (N = 4) | Health facilities (N = 20), Districts (N = 2) | Descriptive statistics on frequency, mean and standard deviation, Cross tabulation to compare performance of the participants |
| 2 | Semi structured interview guides for in-depth interview | All levels (national, regional, district, health facility, community representatives) | face to face | Patients, dispensary (N = 32), Patients, health center (N = 32), Health care workers, Dispensary (N = 16), Health care workers Health centers (N = 16), | Health facility governing committee members (N = 80), CHMT members (N = 60), RHMT members (N = 4), implementing partners (N = 4) | Thematic analysis will be used to analyze qualitative data using ATLAS.ti 8 software |
| 3 | Structured questionnaire | Health facility level | Face to face (guided), in ODK | Health care workers (N = 440), exit patients (N = 422) | Health care workers (N = 200) | Descriptive statistics (mean, standard deviation), Cross tabulation to compare groups' performance Regression analysis for checking statistical significances among and between the participants Paired sample t test for comparing means between the baseline and end-line surveys |
| 4 | DHS2 extraction form | Health facility level / District level | electronic records | Health facilities (N = 40), Districts (N = 4) | | Descriptive statistics on frequency, mean and standard deviation, Cross tabulation to compare performance of health facilities/ districts, Paired sample t-test for comparing means between the baseline and end-line surveys |

Authorship of the resulting papers will be decided based on the International Committee of Medical Journal Editors criteria.

## Discussion

Data use initiatives are among the most complex health system interventions implemented at different levels in the health care settings. The interventions in the course of implementation, required articulation of processes and inputs along multiple levels (district, health facility to individual frontline health care workers), actors and actions that required also support and continuous technical backstopping from different levels and actors.

This study represents a first attempt to document the implementation process, determine the implementation outcomes, understand the determinants of implementation processes and outcomes and determine the effects of data use initiatives on quality of care and health outcomes.

Our implementation-effectiveness hybrid design that takes a before and after controlled design to determine the effects of data use initiatives while also understanding the counterfactual is expected to provide a holistic picture on the interaction between implementation outcomes and program outcomes [24]. The mixed methods design that takes into consideration the implementation process and outcomes amid the influence of contextual factors, is envisaged to provide a comprehensive picture of the influencers of the implementation processes as documented in existing evidence [34].

The current study is designed to yield the best possible evidence to guide implementation of data use initiatives in the primary health care settings. The development of the TOC and monitoring and evaluation framework that saw a combined team of program developers, implementers and researchers, represents a best practice in designing implementation-effectiveness hybrid studies as proposed also in previous evidence in implementation research [35].

The use of mixed-methods approaches as well as multiple tools of collecting data will enable the development of recommendations to improve future implementation of the data use initiatives and other implementation studies of similar nature by identifying trends, challenges and potential solutions to implementation challenges amid contextual influences as well as design issues if they exist. We must also acknowledge the limitations that this study may prone to missing data as well as recall bias as collection of data will take place some months after start of the data use interventions.

Although, we will apply matching in the course of selecting the intervention and control facilities, the fact that we will recruit primary health facilities across the four regions which vary in size, geography, population served, there may be some variations in outcomes. Moreover, although the before–after controlled studies can help overcome the problem of confounding, there is always a risk of unidentified confounders when there is no randomization thus leading to some risk for confounding or bias [36].

This protocol was developed amid the challenges of COVID-19 pandemic that limited face to face meetings. We managed to minimize face to face encounters and relied on virtual communications that could have some effects on same decisions. The various waves of the Covid-19 that presented differently were seen as barriers to face to face meeting and therefore rely on web-based platforms.

## Conclusion

In conclusion, this study is expected to generate evidence on the implementation process and outcomes of the data use interventions in Primary health care settings in Tanzania. As such the findings from this study will enhance understanding of whether and how data use initiatives might improve patients and health systems outcomes amid the influence of contextual factors in the course of implementation. The theoretical approaches and methodologies described in our protocol may be useful in informing the design of future studies on the evaluation of complex interventions in primary health care and beyond.

## Supporting information

**S1 File. Theory of change for deploying data use initiatives in the primary health care in Tanzania.**
(DOCX)

**S2 File. Data collection tools.**
(DOCX)

## Author Contributions

**Conceptualization:** Albino Kalolo, Mackfallen Anasel, Ntuli A. Kapologwe, Stella Mujaya.

**Funding acquisition:** Albino Kalolo.

**Methodology:** Albino Kalolo, Anosisye M. Kesale, Mackfallen Anasel, Ntuli A. Kapologwe, Jennie Jaribu, James T. Kengia.

**Writing – original draft:** Albino Kalolo, Anosisye M. Kesale, Mackfallen Anasel, Ntuli A. Kapologwe, Stella Mujaya, James T. Kengia.

**Writing – review & editing:** Albino Kalolo, Anosisye M. Kesale, Mackfallen Anasel, Ntuli A. Kapologwe, Jennie Jaribu, Stella Mujaya, James T. Kengia.

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
