## [Decision Letter · Decision Letter 0]

7 Sep 2023

PONE-D-22-31731Documenting the implementation processes and effects of the data use initiatives in primary health care settings in Tanzania: A before-after mixed methods study protocolPLOS ONE

Dear Dr. Kalolo,

Thank you for submitting your manuscript to PLOS ONE. After careful consideration, we feel that it has merit but does not fully meet PLOS ONE’s publication criteria as it currently stands. Therefore, we invite you to submit a revised version of the manuscript that addresses the points raised during the review process.

We look forward to receiving your revised manuscript.

Kind regards,

Ermel Johnson, MD, MPH, PhD candidate

Academic Editor

PLOS ONE

“Jennie Jaribu works with UNICEF, Tanzania. Other authors declare no competing interests.”

“The authors wish to thank the teams from Data.Fi, UNICEF, CERHI, PORALG and the councils participating in the data use intervention for their inputs during the preparation of the theory of change. We are also grateful to UNICEF Tanzania for funding this study.”

“Jennie Jaribu works with UNICEF, Tanzania. Other authors declare no competing interests.”

Please include your amended statements within your cover letter; we will change the online submission form on your behalf."

5. We notice that your supplementary files are included in the manuscript file. Please remove them and upload them with the file type 'Supporting Information'. Please ensure that each Supporting Information file has a legend listed in the manuscript after the references list.

Reviewers' comments:

Reviewer's Responses to Questions

**Comments to the Author**

1. Does the manuscript provide a valid rationale for the proposed study, with clearly identified and justified research questions?

Reviewer #1: Yes

Reviewer #2: Yes

2. Is the protocol technically sound and planned in a manner that will lead to a meaningful outcome and allow testing the stated hypotheses?

Reviewer #1: Yes

Reviewer #2: Yes

3. Is the methodology feasible and described in sufficient detail to allow the work to be replicable?

Reviewer #1: Yes

Reviewer #2: Yes

4. Have the authors described where all data underlying the findings will be made available when the study is complete?

Reviewer #1: No

Reviewer #2: Yes

5. Is the manuscript presented in an intelligible fashion and written in standard English?

Reviewer #1: Yes

Reviewer #2: Yes

6. Review Comments to the Author

You may also provide optional suggestions and comments to authors that they might find helpful in planning their study.

Reviewer #1: How was the sample size calculated in the component aiming at establishing the effects of data use initiatives?

Provide a brief description of the data use initiatives so as to allow for understanding what the standard you are measuring the process of implementation against is and provide a reference

Reviewer #2: For research protocols always use futuristic tense "will be", lines 144 and 145

Line 122: "fastness" instead of "fidelity"

Line 125: "unravel" instead of "disentangle"

Line 212: Mention the contextual influences

Line 213: "The study will adopt the following steps" instead of "we will follow the following steps"

7. PLOS authors have the option to publish the peer review history of their article (what does this mean?). If published, this will include your full peer review and any attached files.

Reviewer #1: No

Reviewer #2: **Yes: **Hadiza Mudi

---

## [Author Response · Author response to Decision Letter 0]

8 Oct 2023

Response to reviewers are detailed in a document named " response to reviewers

---

## [Decision Letter · Decision Letter 1]

29 Apr 2024

Documenting the implementation processes and effects of the data use initiatives in primary health care settings in Tanzania: A before-after mixed methods study protocol

PONE-D-22-31731R1

Dear Dr. Kalolo,

We’re pleased to inform you that your manuscript has been judged scientifically suitable for publication and will be formally accepted for publication once it meets all outstanding technical requirements.

Kind regards,

Maryam Afshari

Academic Editor

PLOS ONE

Additional Editor Comments (optional):

Reviewers' comments:

Reviewer's Responses to Questions

**Comments to the Author**

1. Does the manuscript provide a valid rationale for the proposed study, with clearly identified and justified research questions?

Reviewer #2: Yes

Reviewer #3: Yes

2. Is the protocol technically sound and planned in a manner that will lead to a meaningful outcome and allow testing the stated hypotheses?

Reviewer #2: Yes

Reviewer #3: Yes

3. Is the methodology feasible and described in sufficient detail to allow the work to be replicable?

Reviewer #2: Yes

Reviewer #3: Yes

4. Have the authors described where all data underlying the findings will be made available when the study is complete?

Reviewer #2: Yes

Reviewer #3: Yes

5. Is the manuscript presented in an intelligible fashion and written in standard English?

Reviewer #2: Yes

Reviewer #3: Yes

6. Review Comments to the Author

You may also provide optional suggestions and comments to authors that they might find helpful in planning their study.

Reviewer #2: Q4. The dissemination process on how the results from the study will be shared with participating districts and health facilities as well as dissemination through scientific conference presentation and peer reviewed journals have been mentioned (line 412-416) however, Data sharing have been stated not to be applicable to the article as no datasets were generated or analyzed during the study (line 490-492).

Authors have addressed most of the concerns and suggestions earlier raised, they have given satisfactory answers to the questions and has made the necessary revisions.

Article is now in good shape to go for the publication.

Reviewer #3: The authors have adequately addressed the reviewers' comments from the previous round. The changes made have improved the clarity and quality of the manuscript. Here are some additional suggestions:

-The description of the data use initiatives could be expanded slightly to provide more context and details for readers who may not be familiar with the background. For example, roughly how many facilities are involved, when did implementation start, who are the key partners involved etc.

-In the methods section, please clarify the timeline of data collection relative to the initiative implementation. For example, will baseline data be collected before implementation begins? How long after implementation starts will endline data be collected?

-For qualitative data, consider mentioning target sample sizes or when saturation is expected to be reached.

-In the limitations, the authors could comment on risks of contamination between intervention and control arms given they are in the same regions.

-Do double check the referenced figures/tables align with the numbered lists in text.

Overall the study protocol is well designed and described. The authors have been responsive to reviewer feedback. Pending the minor clarifications above, I would recommend accepting this revised manuscript for publication, assuming the other reviewer is satisfied with the changes made. The protocol details an important evaluation that can inform implementation and scale up of data use initiatives.

7. PLOS authors have the option to publish the peer review history of their article (what does this mean?). If published, this will include your full peer review and any attached files.

Reviewer #2: No

Reviewer #3: No

---

## [Editor Report · Acceptance letter]

9 May 2024

PONE-D-22-31731R1 

PLOS ONE

Dear Dr. Kalolo, 

I'm pleased to inform you that your manuscript has been deemed suitable for publication in PLOS ONE. Congratulations! Your manuscript is now being handed over to our production team.

Kind regards, 

on behalf of

Dr. Maryam Afshari 

Academic Editor

PLOS ONE